# OpenReview forum: "CapRL: Stimulating Dense Image Caption Capabilities via Reinforcement Learning"
_ICLR.cc/2026/Conference — ICLR 2026 Poster_

### Official Review · Reviewer_hugc · 2025-10-27

**Soundness:** 2
**Presentation:** 3
**Contribution:** 3
**Rating:** 4
**Confidence:** 3

**Summary:**

This paper proposes CapRL (Captioning Reinforcement Learning), a new framework applying Reinforcement Learning with Verifiable Rewards (RLVR) to the open-ended image captioning task. CapRL defines caption quality in terms of utility: a caption is good if it enables a non-visual LLM to correctly answer multiple-choice VQA questions about the original image. The method uses an LVLM to generate a caption, and a vision-free LLM to answer questions based solely on that caption, with the resulting accuracy serving as a verifiable binary reward. Then, Group Relative Policy Optimization (GRPO) is used to train the LVLM with the reward. To support question creation, the authors develop a specific QA curation pipeline to ensure the quality of questions and answers. The authors introduce a large-scale CapRL-5M dataset generated using CapRL-3B, and demonstrate significant improvements across 12 multimodal benchmarks (InfoVQA, ChartQA, SEED, MMStar, etc.), showing performance comparable to much larger models (e.g., Qwen2.5-VL-72B).

**Strengths:**

The core idea of redefining caption quality through its verifiable downstream utility (i.e., enabling correct VQA responses) is interesting to overcome the long-standing reward-design challenge in open-ended language generation.

The paper is well organized and readable. The methodology is clearly structured, with detailed explanations.

The dataset construction (CapRL-5M) is beneficial to the community if the author makes it open-source.

**Weaknesses:**

The two-stage pipeline (multiple QA samples per image, multiple-choice shuffling, etc.) introduces heavy computational overhead.

The reward signal depends entirely on a single non-visual LLM’s comprehension and question-answering ability. If that LLM is weak, biased, or miscalibrated, CapRL may learn suboptimal behaviors.

**Questions:**

How does CapRL perform when the reward LLM is replaced with models of varying strengths (weaker, stronger models)?

What is the actual computational cost of each stage in the CapRL pipeline?

How sensitive is CapRL to the accuracy of QA filtering? What happens if weakly grounded questions are included?

How does CapRL handle creative or abstract images that do not have factual QAs (e.g., artwork, memes)?

Do the authors open CapRL-5M to the community?

---

> ### Author Response · Authors · 2025-11-21
> **Reponse to Reviewer hugc (1/2)**
>
> We greatly appreciate the reviewer’s insightful comments, careful evaluation, and positive feedback. In response, we have carried out additional experiments and revised the manuscript accordingly. Our point-by-point replies are provided below.
>
> > **W1 & Q2: The two-stage pipeline (multiple QA samples per image, multiple-choice shuffling, etc.) introduces heavy computational overhead.**
>
> We thank the reviewer for raising this important concern regarding computational overhead, as it gives us the opportunity to clarify the actual efficiency of our pipeline.
> To address this, we can decompose the training loop of CapRL into three distinct stages: Rollout, Reward Generation (LLM answering QA based on captions), and Training (Optimization). It is important to note that both the Rollout and Training stages are inherent components of any standard RL framework.
>
> |  | Rollout | QA based on caption | Training | total |
> | :--- | :--- | :--- | :--- | :--- |
> | CapRL | 51.1s | 6.5s | 58.1s | 115.7s |
> | Genaral-RL-Training | 50.5s | 0s | 59.2s | 109.7s |
>
> To rigorously evaluate the time cost, we conducted experiments using 8 GPUs for the Rollout and Training stages, and 2 separate GPUs dedicated to the Reward Generation. As shown in the table, for a single complete RL step, CapRL requires 115.7s, whereas the standard RL baseline requires 109.7s. This represents a marginal increase of only 5.1% in total time.
> The additional overhead is incurred solely during the Reward Generation stage, which is computationally inexpensive in our setup. By deploying Qwen2.5-3B on a vLLM server across the 2 dedicated GPUs, the QA process takes only 6.5s per step. This high efficiency is attributed to two main factors:
>
> - Minimal Generation Length: The model is only required to output a single token (the selected option) for each query.
>
> - Inference Acceleration: The use of vLLM significantly speeds up the serving process.
>
> Therefore, the CapRL algorithm introduces only minimal computational overhead.
>
> > **W2 & Q1: The reward signal depends entirely on a single non-visual LLM’s comprehension and question-answering ability. If that LLM is weak, biased, or miscalibrated, CapRL may learn suboptimal behaviors.**
>
> We thank the reviewer for raising this insightful question regarding answer-model sensitivity. To address this concern, we conducted extensive experiments to investigate the impact of different LLM sizes on the CapRL model.
> Specifically, we employed the Qwen2.5 series as the answer-model, utilizing five different model sizes ranging from 0.5B to 32B to answer questions based on the captions. The results are summarized in the table below.
>
> | Caption Model | ChartQA | SEED | Info VQA | MMStar | ChartQA Pro | MMMU Pro | MMMU | MathVerse | CharXiv | WeMath | MathVision | Average |
> | :--- | :--- | :--- | :--- | :--- | :--- | :--- | :--- | :--- | :--- | :--- | :--- | :--- |
> | Qwen2.5-VL-3B | 65.6 | 64.1 | 40.2 | 46.4 | 27.1 | 28.6 | 35.1 | 32.8 | 21.8 | 54.4 | 22.6 | 39.9 |
> | CapRL-3B-20k-Answer0.5b | 74.9 | 66.8 | 54.9 | 48.6 | 33.7 | 30.1 | 35.8 | 34.3 | 24.7 | 56.1 | 23.5 | 43.9 |
> | CapRL-3B-20k-Answer1.5b | 76.6 | 67.6 | 59.5 | 50.7 | 36.2 | 29.9 | 36.7 | 35.2 | 26.7 | 56.6 | 23.0 | 45.3 |
> | CapRL-3B-20k-Answer3b | 77.1 | 68.1 | 59.6 | 50.8 | 36.9 | 30.0 | 36.5 | 34.8 | 28.3 | 57.7 | 23.4 | 45.7 |
> | CapRL-3B-20k-Answer7b | 77.0 | 67.9 | 60.2 | 51.0 | 35.7 | 30.2 | 36.9 | 34.7 | 29.1 | 56.9 | 23.6 | 45.7 |
> | CapRL-3B-20k-Answer32b | 78.6 | 68.5 | 60.1 | 50.2 | 35.9 | 30.0 | 35.7 | 34.5 | 28.3 | 56.3 | 23.2 | 45.6 |
>
> We observed that even when using the smallest 0.5B model, CapRL still yields a significant improvement in the captioning capability of Qwen2.5-VL-3B; however, there is a noticeable performance drop compared to the 3B model. Performance improves marginally from 1.5B to 3B and then begins to saturate, with no significant variance observed between the 7B and 32B models.
> These findings indicate that the answer-model does not require excessive parameter size; it simply requires a fundamental capability for information extraction. However, extremely small models, such as the 0.5B variant, appear to lack sufficient capacity for effective extraction.
>
> We have added these quantitative comparisons to Appendix for comprehensive LLM ablation.

---

> ### Author Response · Authors · 2025-11-21
> **Reponse to Reviewer hugc (2/2)**
>
> > **Q3: How sensitive is CapRL to the accuracy of QA filtering? What happens if weakly grounded questions are included?**
>
> We thank the reviewer for this insightful question regarding the sensitivity of CapRL to the accuracy of QA filtering and the potential impact of weakly grounded data. To address this, we conducted an additional experiment to evaluate the model's performance when training with lower-quality data.
> We still used QwenVL-72B to generate QA pairs, and then asked QwenVL-3B to answer the questions both with and without seeing the image. For each question, we shuffled the answer choices and obtained the model answer four times. We filtered out QA pairs for which the accuracy with images was below 75% or the accuracy without images was above 75%. These QA pairs formed the Weakly-Ground-20k set, which we compared against the Strict-Ground-20k set obtained using our original strict filtering. The experimental results are shown below.
>
> | Caption Model | ChartQA | SEED | Info VQA | MMStar | ChartQA Pro | MMMU Pro | MMMU | MathVerse | CharXiv | WeMath | MathVision | Average |
> | :--- | :--- | :--- | :--- | :--- | :--- | :--- | :--- | :--- | :--- | :--- | :--- | :--- |
> | Qwen2.5-VL-3B | 65.6 | 64.1 | 40.2 | 46.4 | 27.1 | 28.6 | 35.1 | 32.8 | 21.8 | 54.4 | 22.6 | 39.9 |
> | CapRL-Weak-Ground-20k | 76.3 | 67.2 | 58.1 | 49.5 | 35.9 | 29.6 | 36.2 | 33.5 | 27.1 | 55.3 | 23.4 | 44.7 |
> | CapRL-Strict-Ground-20k | 77.1 | 68.1 | 59.6 | 50.8 | 36.9 | 30.0 | 36.5 | 34.8 | 28.3 | 57.7 | 23.4 | 45.7 |
>
> The results indicate that CapRL-Weak-Ground-20k performs noticeably worse than CapRL-Strict-Ground-20k, but still provides a substantial improvement over Qwen2.5-VL-3B. This demonstrates that CapRL is relatively robust to the accuracy of QA filtering, even when the QA quality is lower, our algorithm can still extract useful signals and significantly enhance the capability of Qwen2.5-VL-3B. However, it is important to note that if one aims to obtain the best-performing CapRL model, applying our strict filtering procedure is necessary.
>
> > **Q4: How does CapRL handle creative or abstract images that do not have factual QAs (e.g., artwork, memes)?**
>
> We thank the reviewer for raising this insightful point regarding the handling of non-factual content. For creative or abstract images, such as artwork, the construction of QAs is highly dependent on the specific target task.
> For instance, if the objective is to enhance the base model's aesthetic understanding or its ability to describe artistic styles, we can leverage closed-source models or domain-specific models to generate relevant Q&A pairs. By constructing questions such as "Which art movement does this work belong to?" or "What is the overall emotion conveyed by the painting?", we can effectively guide the model to improve its capability in appreciating and describing abstract artistic elements.
>
> > **Q5: Do the authors open CapRL-5M to the community?**
>
> We appreciate the reviewer’s concern, which allows us to clarify our plans for release. We are strong advocates for the power of open source and firmly believe that an active community benefits everyone. To that end, we will **release all our training/evaluation code, dataset, and trained models** immediately after the paper is accepted. Our goal is to empower the community to build upon our work in dense image captioning, while also leveraging community feedback to uncover any shortcomings in CapRL and drive its continuous improvement.
>
> Thanks again for your constructive comments and meticulous review, which have helped us significantly improve the paper's quality! Please don't hesitate to let us know if there are any additional clarifications or experiments that we can offer.

---

> ### Author Response · Authors · 2025-11-27
> **Official Comment by Authors**
>
> Dear Reviewer hugc,
>
> We sincerely appreciate your time and effort in providing such meticulous reviews and insightful comments.
>
> Could you please kindly let us know if our rebuttal has addressed your concerns?
>
> If you have any further questions regarding our work, we would be delighted to address them.
>
> Best regards,
>
> Authors

---

### Official Review · Reviewer_UVsw · 2025-10-27

**Soundness:** 2
**Presentation:** 2
**Contribution:** 3
**Rating:** 6
**Confidence:** 4

**Summary:**

This paper introduces CapRL, a framework for synthesizing richer captions to enhance VLM training. Their contributions are two-fold. 1. They incorporate the RLVR paradigm to train CapRL-3B, where the reward is defined by the average score of an LLM answering questions based on the generated captions. 2. They utilize the RL-trained model to produce an additional 3M image-caption pairs, which are combined with ShareGPT4V-1M and DenseFusion-1M to form the CapRL-5M dataset. Experiments show that pretraining on this curated dataset yields consistent performance gains over existing pretraining datasets under the same model architecture.

**Strengths:**

1. The idea of combining RLVR + data synthesis is novel and effective, as demonstrated in the results of Table 1.

2. As mentioned in appendix E, they employ a data filtering pipeline to alleviate the data-leakage issue in their pretraining corpus.

3. The proposed synthesis pipeline is scalable with the model scale being only 3B.

**Weaknesses:**

1. While the paper claims that CapRL reduces hallucination, no quantitative evidence is provided for either CapRL-3B or models pretrained on CapRL-1M/5M. The authors are encouraged to report hallucination metrics such as POPE [1] or CHAIR [2] to support this claim.

2. The experiments in Table 3 do not appear to provide a fully fair comparison. The evaluation relies on text-only captions to generate responses, whereas only CapRL has been RL-tuned specifically to produce detailed captions, giving it an inherent advantage. In real-world applications, models are typically used in an end-to-end manner rather than in such a disjoint caption–response setup. Therefore, weaker performance under this prism evaluation does not necessarily indicate poorer downstream capability. I understand the authors’ intention to demonstrate that higher-quality captions can improve downstream QA performance, but a more direct and convincing evaluation would be to apply CapRL as a post-training stage to a VLM and then compare it with other models on these benchmarks under standard end-to-end settings.

3. In the QA curation process, the authors first use QwenVL-72B to generate QA pairs and then QwenVL-3B to verify that the answers are visually grounded and image-dependent. However, this filtering step may remove examples where the smaller model disagrees with the larger one, potentially biasing the dataset toward easier instances (i.e., those correctly answered by the 3B model) and excluding more challenging or informative pairs that only the 72B model answers correctly.

4. It would be helpful to include captioning benchmarks and quantitative metrics to assess the improvements of CapRL-3B over QwenVL-3B, providing a clearer measure of caption quality.

5. (Minor) The title and abstract may be somewhat misleading, as the paper does not present quantitative analyses of caption quality; instead, the evaluation primarily focuses on downstream task performance when using the constructed datasets for pretraining.

[1] Evaluating Object Hallucination in Large Vision-Language Models, EMNLP 2023.

[2] Object Hallucination in Image Captioning, ACL 2018.

**Questions:**

Please see the above weakness section.

---

> ### Author Response · Authors · 2025-11-21
> **Reponse to Reviewer UVsw (1/3)**
>
> We sincerely appreciate the reviewer’s thoughtful comments, thorough evaluation, and positive feedback. In response, we have carried out additional experiments and revised the manuscript accordingly. Detailed replies are provided below.
>
> > **W1 &W4: While the paper claims that CapRL reduces hallucination, no quantitative evidence is provided for either CapRL-3B or models pretrained on CapRL-1M/5M. The authors are encouraged to report hallucination metrics such as POPE [1] or CHAIR [2] to support this claim.
> It would be helpful to include captioning benchmarks and quantitative metrics to assess the improvements of CapRL-3B over QwenVL-3B, providing a clearer measure of caption quality.**
>
> We thank the reviewer for this constructive suggestion regarding the quantitative evaluation of hallucinations.
> Following your advice, we evaluated the models pretrained on CapRL-1M/5M (under the Qwen2.5-7B + Qwen2.5-ViT setting) using the POPE benchmark. As shown in the table, the models trained with the CapRL dataset achieve significant improvements compared to the Share-1M and Dense-1M baselines.
>
> | Model | Share-1M | Dense-1M | CapRL-1M | CapRL-5M |
> | :--- | :--- | :--- | :--- | :--- |
> | POPE | 87.5 | 87.4 | 88.4 | 88.8 |
>
> Regarding the CHAIR benchmark [2], we carefully considered its inclusion. However, we noted that CHAIR (proposed in 2018) relies on exact matching with a limited, fixed set of object nouns, which causes it to miss many objects and fine-grained details. While effective for the short captions common at the time, this mechanism is less suitable for modern, powerful MLLMs, which tend to generate stylistic and detail-rich captions where simple noun matching may yield inaccurate evaluation. Therefore, to provide a more comprehensive and up-to-date assessment of the captions generated by CapRL, we have instead conducted: (1) evaluations on several Standard Caption Benchmarks, (2) Human Study, and (3) GPT-5 Evaluation.
>
> **(1) Standard Caption Benchmarks**
>
> We conducted extensive evaluations using ALOHa [1], CLAIR [2], and the Factuality and Coverage metrics proposed by CapMAS [3]. These four metrics rely on GPT-based judging or specialized match algorithms to provide automated, quantitative assessments of caption quality. Among them, ALOHa and the Factuality metric are specifically designed to detect hallucinations.
>
> | Caption Model    | ALOHa | CLAIR | Factuality | Coverage | avg  |
> |------------------|-------|-------|------------|----------|------|
> | Qwen2.5-VL-3B     | 62.2  | 68.9  | 79.3       | 59.3     | 67.4 |
> | Qwen2.5-VL-72B    | 66.1  | 80.6  | 83.8       | 69.7     | 75.1 |
> | CapRL-3B          | 65.7  | 80.7  | 80.5       | 73.7     | 75.2 |
>
> Across these metrics, CapRL-3B significantly outperforms Qwen2.5-VL-3B by 7.8% and achieves performance comparable to Qwen2.5-VL-72B. In particular, when it comes to hallucination benchmarks such as ALOHa and Factuality, CapRL-3B clearly outperforms Qwen2.5-VL-3B. Furthermore, CapRL-3B shows a remarkably large advantage on the Coverage metric, exceeding Qwen2.5-VL-3B and Qwen2.5-VL-72B by 14.4% and 4.0%, respectively. These results highlight that our CapRL algorithm substantially unlocks the potential of Qwen2.5-VL-3B, enabling it to capture image details much more comprehensively.
>
> **(2) Human Study**
>
> To further validate these findings, we randomly sampled 200 images from MMStar, ChartQA, InfoVQA, and MMMU, generating captions using Qwen2.5-VL-3B, Qwen2.5-VL-72B, and CapRL-3B. We invites 9 human judges to evaluate the caption quality based on four criteria: **Detail Coverage, Content Accuracy, Expression Fluency, and Succinctness Degree**. To ensure objectivity, we adhered to a strict double-blind protocol where evaluators were randomly assigned captions without knowledge of the source model. The scores for all four dimensions range from 1 to 5.
>
> | Caption Model    | Detail Coverage | Content Accuracy | Expression Fluency | Succinctness Degree | Avg  |
> |------------------|-----------------|------------------|---------------------|----------------------|------|
> | Qwen2.5-VL-3B     | 3.28            | 3.45             | 4.55                | 3.95                 | 3.81 |
> | Qwen2.5-VL-72B    | 4.32            | 4.26             | 4.73                | 4.21                 | 4.38 |
> | CapRL-3B          | 4.68            | 4.16             | 4.65                | 4.15                 | 4.41 |
>
> As shown in the table, CapRL-3B achieves a much higher average score than Qwen2.5-VL-3B and comparable with Qwen2.5-VL-72B. The improvement is especially prominent in Detail Coverage, demonstrating CapRL-3B’s strong ability to capture fine-grained visual information. The results on Expression Fluency indicate that enhancing caption quality via CapRL does not compromise the model’s linguistic fluency. In addition, CapRL-3B exhibits obvious gains over Qwen2.5-VL-3B in Content Accuracy and Succinctness Degree.

---

> ### Author Response · Authors · 2025-11-21
> **Reponse to Reviewer UVsw (2/3)**
>
> **(3) GPT5 Evaluation**
>
> | Caption Model    | Detail Coverage | Content Accuracy | Expression Fluency | Succinctness Degree | Avg  |
> |------------------|-----------------|------------------|---------------------|----------------------|------|
> | Qwen2.5-VL-3B     | 2.85            | 3.08             | 4.30                | 4.02                 | 3.56 |
> | Qwen2.5-VL-72B    | 4.21            | 4.42             | 4.59                | 4.12                 | 4.34 |
> | CapRL-3B          | 4.38            | 4.39             | 4.49                | 4.05                 | 4.33 |
>
> Using the exact same set of images and captions from the Human Study, we employed GPT-5 to score the outputs using a unified prompt. As shown in the table, GPT-5’s scoring trends align closely with the human study, once again confirming the strong captioning ability of CapRL-3B. We also observed that GPT-5 appears stricter than human judges regarding Detail Coverage, especially for Qwen2.5-VL-3B, likely because GPT-5 attends to more fine-grained visual details than human evaluators.
>
> We have (1) added these quantitative comparisons to the experimental and appendix section (2) provided the prompt used for GPT5 in the Appendix.
>
> [1] Petryck et al., "ALOHa: A New Measure for Hallucination in Captioning Models"
>
> [2] Chan et al., "CLAIR: Evaluating Image Captions with Large Language Models"
>
> [3] Lee et al., "Toward Robust Hyper-Detailed Image Captioning: A Multiagent Approach and Dual Evaluation Metrics for Factuality and Coverage"
>
> > **W2: The experiments in Table 3 do not appear to provide a fully fair comparison. The evaluation relies on text-only captions to generate responses, whereas only CapRL has been RL-tuned specifically to produce detailed captions, giving it an inherent advantage. In real-world applications, models are typically used in an end-to-end manner rather than in such a disjoint caption–response setup. Therefore, weaker performance under this prism evaluation does not necessarily indicate poorer downstream capability. I understand the authors’ intention to demonstrate that higher-quality captions can improve downstream QA performance, but a more direct and convincing evaluation would be to apply CapRL as a post-training stage to a VLM and then compare it with other models on these benchmarks under standard end-to-end settings.**
>
> We thank the reviewer for this insightful comment regarding the experimental setup in Table 3. We appreciate the opportunity to clarify the specific objective of this evaluation.
>
> We would like to emphasize that the primary goal of CapRL is to enhance the captioning capabilities of the model. Consequently, all our experiments, including those in Table 1 and Table 3, are designed to assess the quality of captions generated by different models from various perspectives.
> In this context, the primary purpose of Table 3 is not to demonstrate that "higher-quality captions improve downstream QA performance". Instead, we treat this relationship as an established premise to benchmark the quality of captions generated by different models. To achieve this, we adopted a pipeline evaluation where captions generated by different models are fed into a fixed Large Language Model (LLM) for Question Answering.
>
> The rationale behind this proxy metric is straightforward: if a generated caption contains fewer hallucinations and provides richer visual details (indicative of high quality), the LLM will naturally achieve higher accuracy. Conversely, if a caption suffers from severe hallucinations or lacks sufficient detail, the LLM's performance will degrade. Therefore, by fixing the answering module (the LLM), this setup allows us to explicitly compare the caption quality produced by the different baselines.

---

> ### Author Response · Authors · 2025-11-21
> **Reponse to Reviewer UVsw (3/3)**
>
> > **W3: In the QA curation process, the authors first use QwenVL-72B to generate QA pairs and then QwenVL-3B to verify that the answers are visually grounded and image-dependent. However, this filtering step may remove examples where the smaller model disagrees with the larger one, potentially biasing the dataset toward easier instances (i.e., those correctly answered by the 3B model) and excluding more challenging or informative pairs that only the 72B model answers correctly.**
>
> We appreciate the reviewer’s concern regarding the overall difficulty of the QA dataset. And we would like to clarify that this design was intentionally adopted to construct QA pairs of appropriate difficulty for the policy model.
>
> To directly evaluate the effect of training on a more challenging QA dataset, we conducted the following experiment. We continued to use QwenVL-72B to generate QA pairs, then filtered out any items that QwenVL-3B could answer correctly by looking only at the question, so as to avoid leakage. After that, we had QwenVL-3B answer the questions with the image, shuffling the options and answering each question four times. We kept only those QA pairs for which QwenVL-3B’s accuracy was below 50%, forming the Difficult-20k set, which we compared against Ori-20k. The results are shown below.
>
>
> The main reason for this is that for GRPO training, a higher dataset difficulty is not necessarily better. Instead, an appropriate level of difficulty is required for QA (Question Answering) [1,2]. If the difficulty is too high, the rewards across all rollouts will generally be quite low. Consequently, the model will be unable to identify and reward better-performing rollouts, ultimately leading to a decline in model performance. Therefore, the difficulty of the data must be matched to the model's own capabilities when constructing the dataset.
>
> | Caption Model | ChartQA | SEED | Info VQA | MMStar | ChartQA Pro | MMMU Pro | MMMU | MathVerse | CharXiv | WeMath | MathVision | Average |
> | :--- | :--- | :--- | :--- | :--- | :--- | :--- | :--- | :--- | :--- | :--- | :--- | :--- |
> | Qwen2.5-VL-3B | 65.6 | 64.1 | 40.2 | 46.4 | 27.1 | 28.6 | 35.1 | 32.8 | 21.8 | 54.4 | 22.6 | 39.9 |
> | CapRL-Difficult-20k | 74.5 | 67.5 | 59.5 | 50.3 | 35.8 | 29.9 | 35.8 | 33.6 | 27.1 | 56.5 | 22.8 | 44.8 |
> | CapRL-Ori-20k | 77.1 | 68.1 | 59.6 | 50.8 | 36.9 | 30.0 | 36.5 | 34.8 | 28.3 | 57.7 | 23.4 | 45.7 |
>
> [1]"CoDaPO: Confidence and Difficulty-Adaptive Policy Optimization"
>
> [2]"Scaf-GRPO: Scaffolded Group Relative Policy Optimization for Enhancing LLM Reasoning"
>
> > **W5: The title and abstract may be somewhat misleading, as the paper does not present quantitative analyses of caption quality; instead, the evaluation primarily focuses on downstream task performance when using the constructed datasets for pretraining.**
>
> We sincerely thank the reviewer for pointing out the potential ambiguity in our work. To address this, we have carefully refined the phrasing in the abstract to prevent any misunderstanding regarding our contributions. Furthermore, we have incorporated additional quantitative analyses of caption quality within the Experiments section to provide a more comprehensive evaluation.
>
> Thanks again for your constructive comments and meticulous review, which have helped us significantly improve the paper's quality! Please don't hesitate to let us know if there are any additional clarifications or experiments that we can offer.

---

> ### Author Response · Authors · 2025-11-27
> **Official Comment by Authors**
>
> Dear Reviewer UVsw,
>
> We sincerely appreciate your time and effort in providing such meticulous reviews and insightful comments.
>
> Could you please kindly let us know if our rebuttal has addressed your concerns?
>
> If you have any further questions regarding our work, we would be delighted to address them.
>
> Best regards,
>
> Authors

---

### Official Review · Reviewer_MPnM · 2025-11-01

**Soundness:** 3
**Presentation:** 2
**Contribution:** 3
**Rating:** 4
**Confidence:** 3

**Summary:**

The paper proposes CapRL, a reinforcement-learning framework for vision-language models that tackles the problem of reward hacking in open-ended image captioning. Instead of letting a reward model or llm judege directly score captions (which the authors show quickly collapses to trivial, short outputs), CapRL splits evaluation into two stages: (1) use a strong VL model to generate multiple vision-grounded MCQs about an image, and (2) judge a caption by asking a (cheaper) LLM to answer those MCQs using only the caption, averaging over shuffled options to reduce bias. This verifiable reward is then used to RL-tune Qwen2.5-VL variants, and the authors report that the resulting models improve across a wide suite of benchmarks.

**Strengths:**

* The idea of evaluating the generated caption, not in terms of the caption itself, but in terms of how well it provides information for downstream QA seems very interesting and novel.

* More QA data leading to better performance shows the scalability of the method.

**Weaknesses:**

[minor] In Figure 1 of introduction, the authors use the term 'Unified Reward Model as a Judge', but its not exactly clear what this is. Is it an open-source reward model trained to evaluate captionings?

* In Tables 1 and 2 the authors report results on 12 LVLM benchmarks (InfoVQA, DocVQA, ChartQA, RealWorldQA, MathVista, SEED-Bench 2 Plus, MME, RW MMB, MMStar, MMVet, AI2D, GQA). Could the authors clarify what metric is being reported for each dataset (e.g., accuracy / exact match / task-specific score)? Since the main task in the paper is captioning, it would help to state explicitly how these VQA-style evaluations are computed.

* When the authors mention Qwen2.5-3B + Qwen2.5-ViT, does this mean Qwen2.5-VL-3B? If not, what is the difference?

* In the proposed 2-stage reward, the caption is judged by whether an LLM can answer MCQs based solely on that caption. Could you clarify how well this proxy aligns with human-oriented caption quality (fluency, succinctness, informativeness)? In other words, is there a risk that the model is being optimized for captions that are good for downstream QA rather than captions that are good for people, and if so, can the authors provide human evals or qualitative examples to show the proxy is realistic?

**Questions:**

See weaknesses above.

---

> ### Author Response · Authors · 2025-11-21
> **Reponse to Reviewer MPnM (1/3)**
>
> We are truly grateful to the reviewer for the valuable comments, thorough evaluation, and encouraging remarks. In response, we have carried out additional experiments and revised the manuscript accordingly. Our detailed replies are presented below.
>
> > **W1: In Figure 1 of introduction, the authors use the term 'Unified Reward Model as a Judge', but its not exactly clear what this is. Is it an open-source reward model trained to evaluate captionings?**
>
> We thank the reviewer for this insightful comment and for giving us the opportunity to clarify our experimental setting. The term "Unified Reward Model" refers to the work by [1], which is a widely adopted model trained to provide reward signals for both image understanding and image generation tasks. This work has shown that jointly learning to assess multiple tasks may foster a synergistic effect, thereby significantly enhancing the reward model's performance. Due to its remarkable ability to provide rewards, we employed this model to evaluate and reward the quality of captions as a valuable baseline.
> We have explicitly added the explanation of "Unified Reward Model as a Judge" in the Introduction to avoid future confusion.
>
> [1] "Unified Reward Model for Multimodal Understanding and Generation", arXiv:2503.05236.
>
> > **W2: In Tables 1 and 2 the authors report results on 12 LVLM benchmarks (InfoVQA, DocVQA, ChartQA, RealWorldQA, MathVista, SEED-Bench 2 Plus, MME, RW MMB, MMStar, MMVet, AI2D, GQA). Could the authors clarify what metric is being reported for each dataset (e.g., accuracy / exact match / task-specific score)? Since the main task in the paper is captioning, it would help to state explicitly how these VQA-style evaluations are computed.**
>
> We appreciate the reviewer raising this important point, which allows us to provide further clarity on our evaluation setting. To address the confusion, we would like to clarify that we do not use the generated captions to directly answer the questions in these VQA benchmarks.
> Instead, we evaluate the quality of our caption dataset by integrating it into a standard LVLM training pipeline (Pre-training -> SFT). In our experiments, we substitute the caption dataset(ShareGPT-1M, DenseFusion-1M, and CapRL-1M) used during the pre-training stage  while keeping the subsequent Supervised Fine-Tuning (SFT) stage identical for all models. We then evaluate the resulting LVLMs on general VQA benchmarks (e.g., InfoVQA, DocVQA, ChartQA).
>
> The underlying logic for Tables 1 and 2 is based on the widely established consensus[1,2,3] in LVLM research: higher-quality captions during the pre-training stage lead to superior model performance after consistent SFT. Therefore, the results reported are the performance metrics of the models trained on these datasets, serving as a direct proxy for the quality of the caption data itself.
>
> [1]"ShareGPT4V: Improving Large Multi-Modal Models with Better Captions." arXiv:2311.12793
>
> [2]"LLaVA-OneVision-1.5: Fully Open Framework for Democratized Multimodal Training." arXiv:2509.23661
>
> [3]"DenseFusion-1M: Merging Vision Experts for Comprehensive Multimodal Perception" arXiv:2407.08303
>
> > **W3: When the authors mention Qwen2.5-3B + Qwen2.5-ViT, does this mean Qwen2.5-VL-3B? If not, what is the difference?**
>
> We appreciate the reviewer for seeking clarification on our model architecture. To clarify, "Qwen2.5-3B + Qwen2.5-ViT" is not equivalent to Qwen2.5-VL-3B. Instead, it denotes a specific initialization strategy where we combine Qwen2.5-3B-Instruct (a text-pretrained LLM, distinct from the aligned LLM component in Qwen2.5-VL-3B) with the Vision Transformer (ViT) from Qwen2.5-VL-3B. Crucially, the MLP projector connecting them is randomly initialized.
>
> We adopted this setup to strictly simulate a realistic LVLM training scenario. A standard training pipeline typically consists of three stages: Initial Alignment, Further Pre-training, and Supervised Fine-Tuning (SFT). This process necessitates starting with a text-only LLM (which has not yet undergone visual alignment) and a fresh MLP module, rather than using an already fully aligned model like Qwen2.5-VL-3B.
>
> We further clarified these details in the "Implementation Details" part of the Experiment Section.

---

> ### Author Response · Authors · 2025-11-21
> **Reponse to Reviewer MPnM (2/3)**
>
> > **W4: In the proposed 2-stage reward, the caption is judged by whether an LLM can answer MCQs based solely on that caption. Could you clarify how well this proxy aligns with human-oriented caption quality (fluency, succinctness, informativeness)? In other words, is there a risk that the model is being optimized for captions that are good for downstream QA rather than captions that are good for people, and if so, can the authors provide human evals or qualitative examples to show the proxy is realistic?**
>
> We thank the reviewer for this insightful comment regarding the alignment between our proxy reward and human-oriented caption quality. We agree that it is crucial to ensure the captions are not merely overfitting to VQA tasks at the expense of readability. To verify whether models trained by CapRL  meet human standards (e.g., fluency, succinctness, informativeness), we conducted extensive validation experiments. Specifically, we performed (1) Human Study and (2) GPT-5 Evaluation. Furthermore, to provide a more comprehensive assessment of overall quality, we also conducted (3) Caption Benchmark Evaluations.
>
> **(1) Human Study**
>
> To further validate these findings, we randomly sampled 200 images from MMStar, ChartQA, InfoVQA, and MMMU, generating captions using Qwen2.5-VL-3B, Qwen2.5-VL-72B, and CapRL-3B. We invites 9 human judges to evaluate the caption quality based on four criteria: **Detail Coverage, Content Accuracy, Expression Fluency, and Succinctness Degree**. To ensure objectivity, we adhered to a strict double-blind protocol where evaluators were randomly assigned captions without knowledge of the source model. The scores for all four dimensions range from 1 to 5.
>
> | Caption Model    | Detail Coverage | Content Accuracy | Expression Fluency | Succinctness Degree | Avg  |
> |------------------|-----------------|------------------|---------------------|----------------------|------|
> | Qwen2.5-VL-3B     | 3.28            | 3.45             | 4.55                | 3.95                 | 3.81 |
> | Qwen2.5-VL-72B    | 4.32            | 4.26             | 4.73                | 4.21                 | 4.38 |
> | CapRL-3B          | 4.68            | 4.16             | 4.65                | 4.15                 | 4.41 |
>
> As shown in the table, CapRL-3B achieves a much higher average score than Qwen2.5-VL-3B and comparable with Qwen2.5-VL-72B. The improvement is especially prominent in Detail Coverage, demonstrating CapRL-3B’s strong ability to capture fine-grained visual information. The results on Expression Fluency indicate that enhancing caption quality via CapRL does not compromise the model’s linguistic fluency. In addition, CapRL-3B exhibits obvious gains over Qwen2.5-VL-3B in Content Accuracy and Succinctness Degree.
>
> **(2) GPT5 Evaluation**
>
> | Caption Model    | Detail Coverage | Content Accuracy | Expression Fluency | Succinctness Degree | Avg  |
> |------------------|-----------------|------------------|---------------------|----------------------|------|
> | Qwen2.5-VL-3B     | 2.85            | 3.08             | 4.30                | 4.02                 | 3.56 |
> | Qwen2.5-VL-72B    | 4.21            | 4.42             | 4.59                | 4.12                 | 4.34 |
> | CapRL-3B          | 4.38            | 4.39             | 4.49                | 4.05                 | 4.33 |
>
> Using the exact same set of images and captions from the Human Study, we employed GPT-5 to score the outputs using a unified prompt. As shown in the table, GPT-5’s scoring trends align closely with the human study, once again confirming the strong captioning ability of CapRL-3B. We also observed that GPT-5 appears stricter than human judges regarding Detail Coverage, especially for Qwen2.5-VL-3B, likely because GPT-5 attends to more fine-grained visual details than human evaluators.
>
> **(3) Caption Benchmark Evaluation**
>
> We conducted extensive evaluations using ALOHa [1], CLAIR [2], and the Factuality and Coverage metrics proposed by CapMAS [3]. These four metrics rely on GPT-based judging or specialized match algorithms to provide automated, quantitative assessments of caption quality.
>
> | Caption Model    | ALOHa | CLAIR | Factuality | Coverage | avg  |
> |------------------|-------|-------|------------|----------|------|
> | Qwen2.5-VL-3B     | 62.2  | 68.9  | 79.3       | 59.3     | 67.4 |
> | Qwen2.5-VL-72B    | 66.1  | 80.6  | 83.8       | 69.7     | 75.1 |
> | CapRL-3B          | 65.7  | 80.7  | 80.5       | 73.7     | 75.2 |
>
> Across these metrics, CapRL-3B significantly outperforms Qwen2.5-VL-3B by **7.8%** and achieves performance comparable to Qwen2.5-VL-72B. Notably, CapRL-3B shows a particularly large advantage on the Coverage metric, exceeding Qwen2.5-VL-3B and Qwen2.5-VL-72B by **14.4% and 4.0%**, respectively. These results highlight that our CapRL algorithm substantially unlocks the potential of Qwen2.5-VL-3B, enabling it to capture image details much more comprehensively.

---

> ### Author Response · Authors · 2025-11-21
> **Reponse to Reviewer MPnM (3/3)**
>
> We have (1) added these quantitative comparisons to the experimental and appendix section (2) provided the prompt used for GPT5 in the Appendix.
>
> [1] Petryck et al., "ALOHa: A New Measure for Hallucination in Captioning Models"
>
> [2] Chan et al., "CLAIR: Evaluating Image Captions with Large Language Models"
>
> [3] Lee et al., "Toward Robust Hyper-Detailed Image Captioning: A Multiagent Approach and Dual Evaluation Metrics for Factuality and Coverage"
>
> Thanks again for your constructive comments and meticulous review, which have helped us significantly improve the paper's quality! Please don't hesitate to let us know if there are any additional clarifications or experiments that we can offer.

---

> ### Author Response · Authors · 2025-11-27
> **Official Comment by Authors**
>
> Dear Reviewer MPnM,
>
> We sincerely appreciate your time and effort in providing such meticulous reviews and insightful comments.
>
> Could you please kindly let us know if our rebuttal has addressed your concerns?
>
> If you have any further questions regarding our work, we would be delighted to address them.
>
> Best regards,
>
> Authors

---

### Official Review · Reviewer_ML2d · 2025-11-01

**Soundness:** 3
**Presentation:** 3
**Contribution:** 2
**Rating:** 4
**Confidence:** 4

**Summary:**

This paper proposes the CapRL reframeing the captioning task as “utility for QA.” A caption is good if a vision-free LLM can answer the questions about the image using only that caption. The captioner is trained with RL using this verifiable reward (QA accuracy) in a decoupled two-stage pipeline. They start from a small LVLM (Qwen2.5-VL-3B), optimize it with their RL scheme to get CapRL-3B, and then mass-caption images to create a 5M-caption dataset used in standard three-stage LVLM pretraining.

**Strengths:**

- Using QA accuracy as a verifiable signal rather than judge models or reference captions is well-motivated and an interesting way to improve caption quality.
- The proposed approach shows a 3B CapRL-trained captioner matching a **72B** VL model in the decoupled QA-from-caption setting.
- The paper provides useful analyses on (i) training-dataset size, (ii) the number of GRPO generations per step, and (iii) the number of QA pairs per image.

**Weaknesses:**

- Could the method incentivize VQA-specific cues rather than balanced, detailed captions? Please report caption quality directly (standard automatic metrics and a small human study) to show quantitative gains.
- For a fair comparison, evaluate methods on the exact same image set, varying only the training approach. One approach can be evaluating a GRPO-trained VQA model that generates description → reasoning → answer on your QA dataset as a direct comparison.
- The paper measures caption utility only via MCQs. Does the approach transfer to open-ended QA (and related tasks like grounding or summarization)?
- It would be beneficial to include the computation required to generate the QA data, produce captions at scale, and run QA (GPU type/count, GPU-hours, throughput, wall-clock time).
- I wonder if there is any potential bias introduced by using the 72B model to generate the QA dataset used for training.
- Answer-model sensitivity: How much do results change when the answer LLM is swapped (different sizes/models)?

**Questions:**

See above

---

> ### Author Response · Authors · 2025-11-21
> **Reponse to Reviewer ML2d (1/4)**
>
> We sincerely thank the reviewer for the constructive comments, meticulous review and encouraging feedback.
> And we have conducted additional experiments and updated the manuscript accordingly. Our detailed responses are provided below.
>
> > **W1: Could the method incentivize VQA-specific cues rather than balanced, detailed captions? Please report caption quality directly (standard automatic metrics and a small human study) to show quantitative gains.**
>
> We thank the reviewer for this insightful comment.  To rigorously evaluate the quality and balance of the generated captions, we have conducted three additional sets of experiments: (1) Caption Benchmark Evaluation, (2) Human Study, and (3) GPT-5 Evaluation.
>
> **(1) Caption Benchmark Evaluation**
>
> We conducted extensive evaluations using ALOHa [1], CLAIR [2], and the Factuality and Coverage metrics proposed by CapMAS [3]. These four metrics rely on GPT-based judging or specialized match algorithms to provide automated, quantitative assessments of caption quality.
>
> | Caption Model    | ALOHa | CLAIR | Factuality | Coverage | avg  |
> |------------------|-------|-------|------------|----------|------|
> | Qwen2.5-VL-3B     | 62.2  | 68.9  | 79.3       | 59.3     | 67.4 |
> | Qwen2.5-VL-72B    | 66.1  | 80.6  | 83.8       | 69.7     | 75.1 |
> | CapRL-3B          | 65.7  | 80.7  | 80.5       | 73.7     | 75.2 |
>
> Across these metrics, CapRL-3B significantly outperforms Qwen2.5-VL-3B by **7.8%** and achieves performance comparable to Qwen2.5-VL-72B. Notably, CapRL-3B shows a particularly large advantage on the Coverage metric, exceeding Qwen2.5-VL-3B and Qwen2.5-VL-72B by **14.4% and 4.0%**, respectively. These results highlight that our CapRL algorithm substantially unlocks the potential of Qwen2.5-VL-3B, enabling it to capture image details much more comprehensively.
>
> **(2) Human Study**
>
> To further validate these findings, we randomly sampled 200 images from MMStar, ChartQA, InfoVQA, and MMMU, generating captions using Qwen2.5-VL-3B, Qwen2.5-VL-72B, and CapRL-3B. We invites 9 human judges to evaluate the caption quality based on four criteria: **Detail Coverage, Content Accuracy, Expression Fluency, and Succinctness Degree**. To ensure objectivity, we adhered to a strict double-blind protocol where evaluators were randomly assigned captions without knowledge of the source model. The scores for all four dimensions range from 1 to 5.
>
> | Caption Model    | Detail Coverage | Content Accuracy | Expression Fluency | Succinctness Degree | Avg  |
> |------------------|-----------------|------------------|---------------------|----------------------|------|
> | Qwen2.5-VL-3B     | 3.28            | 3.45             | 4.55                | 3.95                 | 3.81 |
> | Qwen2.5-VL-72B    | 4.32            | 4.26             | 4.73                | 4.21                 | 4.38 |
> | CapRL-3B          | 4.68            | 4.16             | 4.65                | 4.15                 | 4.41 |
>
> As shown in the table, CapRL-3B achieves a much higher average score than Qwen2.5-VL-3B and comparable with Qwen2.5-VL-72B. The improvement is especially prominent in Detail Coverage, demonstrating CapRL-3B’s strong ability to capture fine-grained visual information. The results on Expression Fluency indicate that enhancing caption quality via CapRL does not compromise the model’s linguistic fluency. In addition, CapRL-3B exhibits obvious gains over Qwen2.5-VL-3B in Content Accuracy and Succinctness Degree.
>
> **(3) GPT5 Evaluation**
>
> | Caption Model    | Detail Coverage | Content Accuracy | Expression Fluency | Succinctness Degree | Avg  |
> |------------------|-----------------|------------------|---------------------|----------------------|------|
> | Qwen2.5-VL-3B     | 2.85            | 3.08             | 4.30                | 4.02                 | 3.56 |
> | Qwen2.5-VL-72B    | 4.21            | 4.42             | 4.59                | 4.12                 | 4.34 |
> | CapRL-3B          | 4.38            | 4.39             | 4.49                | 4.05                 | 4.33 |
>
> Using the exact same set of images and captions from the Human Study, we employed GPT-5 to score the outputs using a unified prompt. As shown in the table, GPT-5’s scoring trends align closely with the human study, once again confirming the strong captioning ability of CapRL-3B. We also observed that GPT-5 appears stricter than human judges regarding Detail Coverage, especially for Qwen2.5-VL-3B, likely because GPT-5 attends to more fine-grained visual details than human evaluators.
>
> We have (1) added these quantitative comparisons to the experimental and appendix section (2) provided the prompt used for GPT5 in the Appendix.
>
> [1] Petryck et al., "ALOHa: A New Measure for Hallucination in Captioning Models"
>
> [2] Chan et al., "CLAIR: Evaluating Image Captions with Large Language Models"
>
> [3] Lee et al., "Toward Robust Hyper-Detailed Image Captioning: A Multiagent Approach and Dual Evaluation Metrics for Factuality and Coverage"

---

> ### Author Response · Authors · 2025-11-21
> **Reponse to Reviewer ML2d (2/4)**
>
> > **W2: For a fair comparison, evaluate methods on the exact same image set, varying only the training approach. One approach can be evaluating a GRPO-trained VQA model that generates description → reasoning → answer on your QA dataset as a direct comparison.**
>
> We appreciate the reviewer's insightful suggestion to conduct a direct comparison using a GRPO-trained VQA model.
> To enforce this structure, we incorporated a Format Reward during the RL training process on the 20k dataset, ensuring the output adheres to the format: \<caption\>...\</caption\>\<think\>...\</think\>\<answer\>...\</answer\>.  After training with this format, we obtained Reasoning-3B-20k.
> The results are presented in the table below:
>
> | Caption Model     | ChartQA | SEED | InfoVQA | MMStar | ChartQAPro | MMMUPro | MMMU | MathVerse | CharXiv | WeMath | MathVision | Average |
> |-------------------|---------|------|---------|--------|-------------|----------|-------|-----------|---------|--------|------------|---------|
> | Qwen2.5-VL-3B     | 65.6    | 64.1 | 40.2    | 46.4   | 27.1        | 28.6     | 35.1  | 32.8      | 21.8    | 54.4   | 22.6       | 39.9    |
> | Reasoning-3B-20k  | 70.8    | 63.7 | 46.2    | 47.2   | 32.7        | 29.2     | 36.0  | 33.7      | 22.3    | 56.1   | 23.3       | 41.9    |
> | CapRL-3B-20k      | 77.1    | 68.1 | 59.6    | 50.8   | 36.9        | 30.0     | 36.5  | 34.8      | 28.3    | 57.7   | 23.4       | 45.7    |
>
> As shown in the results, while Reasoning-3B-20k achieves a noticeable improvement over Qwen2.5-VL-3B, it falls significantly short of CapRL-3B. Specifically, CapRL-3B surpasses Reasoning-3B-20k by an average of 3.8% across all 11 benchmarks. This substantial margin highlights the superiority of the CapRL algorithm, demonstrating that our method effectively activates the model's captioning capabilities through sparse QA supervision applied directly to the caption.
> Furthermore, we observed during training that the Format Reward for Reasoning-3B-20k saturated rapidly, indicating that the model successfully learned to follow the required structure. We attribute its suboptimal performance to the fact that when answering Multiple Choice Questions (MCQs), the model relies primarily on direct visual information from the image rather than the generated caption. Consequently, the QA task fails to provide effective supervision for the captioning process, limiting the improvement of the model's captioning ability.
> We have included this experiment in the Appendix to provide a comprehensive comparison.
>
> > **W3: The paper measures caption utility only via MCQs. Does the approach transfer to open-ended QA (and related tasks like grounding or summarization)?**
>
> We sincerely thank the reviewer for this insightful suggestion. We fully agree that investigating the transferability of our approach to broader question formats and tasks is a highly promising direction.
> Following your advice, we conducted additional experiments on an open-ended QA format. Specifically, we removed the candidate options from all MCQs, transforming them into questions without fixed standard answers. While this significantly increased the difficulty of verifying the model's responses (as simple string matching is no longer viable), we adopted a reliable solution to address this. We employed a two-step strategy: first, the LLM answers the open-ended question based on the generated caption; second, we construct a prompt containing the generated response, the Ground Truth (GT), and the original question, and ask the LLM to judge the correctness of the answer. We found that this "LLM-as-a-judge" strategy effectively resolved the verification challenge. After training with curated 20k Open-ended QA data, we obtained Open-End-QA-20k.
>
> | Caption Model     | ChartQA | SEED | InfoVQA | MMStar | ChartQAPro | MMMUPro | MMMU | MathVerse | CharXiv | WeMath | MathVision | Average |
> |-------------------|---------|------|---------|--------|-------------|----------|-------|-----------|---------|--------|------------|---------|
> | Qwen2.5-VL-3B     | 65.6    | 64.1 | 40.2    | 46.4   | 27.1        | 28.6     | 35.1  | 32.8      | 21.8    | 54.4   | 22.6       | 39.9    |
> | Open-End-QA-20k   | 78.4    | 67.0 | 60.2    | 50.1   | 36.2        | 30.1     | 36.1  | 34.7      | 27.2    | 56.8   | 23.3       | 45.5    |
> | CapRL-3B-20k      | 77.1    | 68.1 | 59.6    | 50.8   | 36.9        | 30.0     | 36.5  | 34.8      | 28.3    | 57.7   | 23.4       | 45.7    |
>
>
> The results indicate that our CapRL strategy successfully transfers to open-ended QA tasks, achieving performance comparable to the MCQ format across all 11 benchmarks. This outcome is reasonable, as the supervision signals stem from the same QA dataset, implying that the overall optimization direction and capability ceilings are similar across both formats.

---

> ### Author Response · Authors · 2025-11-21
> **Reponse to Reviewer ML2d (3/4)**
>
> Regarding the application of CapRL to related tasks like summarization, while conducting extensive new experiments in this area is beyond the scope of the current paper, we believe the core philosophy of CapRL is directly transferable. For instance, if the goal is to ensure a summary captures key elements (such as time, location, or characters), one could construct MCQs regarding these entities based on the original text. During the RL process, the LLM would then be required to answer these questions based on the generated summary to guide optimization.
>
> > **W4: It would be beneficial to include the computation required to generate the QA data, produce captions at scale, and run QA (GPU type/count, GPU-hours, throughput, wall-clock time).**
>
> We appreciate the reviewer’s suggestion to provide a detailed analysis of the computational resources and efficiency. In response, we have included the specific computation costs for data generation, captioning, and the QA pipeline below.
> All experiments were conducted using NVIDIA A100 GPUs. The computational breakdown is as follows:
>
> (1) **Data Construction and Caption Generation Efficiency**: Using 8 A100 GPUs, we were able to construct 75k QA pairs in approximately 5.5 hours. Furthermore, generating 1 million captions using CapRL-3B took about 10 hours. This demonstrates that the computational overhead for both QA construction and large-scale caption generation is highly efficient and manageable.
>
> |   stage   | Number of GPUs | Time  | Avg Prompt Throughput | Avg Generation Throughput |
> |--------------|----------------|-------|-------------------------|----------------------------|
> | 75k QA data  | 8              | 5.5h  | 2686.9 tokens/s         | 777.5 tokens/s             |
> | CapRL-1M     | 8              | 10.2h | 9130.2 tokens/s         | 2558.0 tokens/s            |
>
> (2) **Training Pipeline and Runtime Analysis**: We also analyzed the time required for a single RL step during CapRL training. We utilized 8 GPUs for rollout and training, with an additional 2 GPUs serving as a vLLM server to handle the QA-based reward calculation (using Qwen2.5-3B).
>
> A single training step can be divided into three stages: Rollout, LLM QA Answering (Reward), and Training. Stages 1 and 3 are standard components of general RL training. As shown in the table, the total time per step for CapRL is 115.7s, compared to 109.7s for General RL Training (which excludes the QA stage). This represents a marginal increase of **only 5.1%**.
>
> | | Rollout | QA based on caption | Training | total |
> |--- | --- | --- | --- | ---|
> | CapRL | 51.1s | 6.5s | 58.1s | 115.7s |
> | Genaral-RL-Training | 50.5s | 0s | 59.2s | 109.7s |
>
> The additional time is solely attributable to the 'LLM QA Answering' stage, which is computationally inexpensive in terms of both time and GPU resources. By deploying a vLLM server on 2 GPUs, answering all questions within a step takes only 6.5s. This high speed is achieved because: (1) Qwen2.5-3B needs to generate only a single token (the selected option) for each query, and (2) vLLM significantly accelerates the inference process. Therefore, the CapRL algorithm introduces negligible temporal overhead compared to standard baselines.
>
> > **W5: I wonder if there is any potential bias introduced by using the 72B model to generate the QA dataset used for training.**
>
> We appreciate the reviewer for raising this valid concern regarding potential bias introduced by the data generation model. To address this quantitatively, we conducted an additional experiment to explore this question.
>
> Specifically, we constructed another QA dataset using a diverse set of models, including InternVL3.5-38B, Qwen2.5VL-72B, and Qwen3VL-32B, and this dataset undergoes the same filtering process with Qwen2.5VL-3B. After training with this dataset, we obtained CapRL-Multi-Model-20k. We hypothesized that if the Qwen2.5VL-72B model induced significant bias, CapRL-Multi-Model-20k would perform much better than CapRL-Ori-20k. The training results are presented below:
>
> | Caption Model | ChartQA | SEED | InfoVQA | MMStar | ChartQAPro | MMMUPro | MMMU | MathVerse | CharXiv | WeMath | MathVision | Average |
> | :--- | :--- | :--- | :--- | :--- | :--- | :--- | :--- | :--- | :--- | :--- | :--- | :--- |
> | Qwen2.5-VL-3B | 65.6 | 64.1 | 40.2 | 46.4 | 27.1 | 28.6 | 35.1 | 32.8 | 21.8 | 54.4 | 22.6 | 39.9 |
> | Multi-Model-20k | 76.7 | 68.6 | 60.5 | 49.9 | 36.8 | 29.8 | 35.7 | 34.5 | 28.5 | 56.5 | 23.8 | 45.6 |
> | Ori-20k | 77.1 | 68.1 | 59.6 | 50.8 | 36.9 | 30.0 | 36.5 | 34.8 | 28.3 | 57.7 | 23.4 | 45.7 |
>
> The results demonstrate that the models trained on the two datasets exhibit negligible differences in performance. This indicates that Qwen2.5VL-72B did not introduce significant bias during dataset construction; consequently, the trained model does not suffer from any noticeable performance degradation.

---

> ### Author Response · Authors · 2025-11-21
> **Reponse to Reviewer ML2d (4/4)**
>
> > **W6: Answer-model sensitivity: How much do results change when the answer LLM is swapped (different sizes/models)?**
>
> We thank the reviewer for raising this insightful question regarding answer-model sensitivity. To address this concern, we conducted extensive experiments to investigate the impact of different LLM sizes on the CapRL model.
> Specifically, we employed the Qwen2.5 series as the answer-model, utilizing five different model sizes ranging from 0.5B to 32B to answer questions based on the captions. The results are summarized in the table below.
>
> | Caption Model | ChartQA | SEED | InfoVQA | MMStar | ChartQAPro | MMMUPro | MMMU | MathVerse | CharXiv | WeMath | MathVision | Average |
> | :--- | :--- | :--- | :--- | :--- | :--- | :--- | :--- | :--- | :--- | :--- | :--- | :--- |
> | Qwen2.5-VL-3B | 65.6 | 64.1 | 40.2 | 46.4 | 27.1 | 28.6 | 35.1 | 32.8 | 21.8 | 54.4 | 22.6 | 39.9 |
> | CapRL-3B-20k-Answer0.5b | 74.9 | 66.8 | 54.9 | 48.6 | 33.7 | 30.1 | 35.8 | 34.3 | 24.7 | 56.1 | 23.5 | 43.9 |
> | CapRL-3B-20k-Answer1.5b | 76.6 | 67.6 | 59.5 | 50.7 | 36.2 | 29.9 | 36.7 | 35.2 | 26.7 | 56.6 | 23.0 | 45.3 |
> | CapRL-3B-20k-Answer3b | 77.1 | 68.1 | 59.6 | 50.8 | 36.9 | 30.0 | 36.5 | 34.8 | 28.3 | 57.7 | 23.4 | 45.7 |
> | CapRL-3B-20k-Answer7b | 77.0 | 67.9 | 60.2 | 51.0 | 35.7 | 30.2 | 36.9 | 34.7 | 29.1 | 56.9 | 23.6 | 45.7 |
> | CapRL-3B-20k-Answer32b | 78.6 | 68.5 | 60.1 | 50.2 | 35.9 | 30.0 | 35.7 | 34.5 | 28.3 | 56.3 | 23.2 | 45.6 |
>
> We observed that even when using the smallest 0.5B model, CapRL still yields a significant improvement in the captioning capability of Qwen2.5-VL-3B; however, there is a noticeable performance drop compared to the 3B model. Performance improves marginally from 1.5B to 3B and then begins to saturate, with no significant variance observed between the 7B and 32B models.
> These findings indicate that the answer-model does not require excessive parameter size; it simply requires a fundamental capability for information extraction. However, extremely small models, such as the 0.5B variant, appear to lack sufficient capacity for effective extraction.
>
> We have added these quantitative comparisons to Appendix for comprehensive LLM ablation.
>
>
> Thanks again for your constructive comments and meticulous review, which have helped us significantly improve the paper's quality! Please don't hesitate to let us know if there are any additional clarifications or experiments that we can offer.

---

> ### Author Response · Authors · 2025-11-27
> **Official Comment by Authors**
>
> Dear Reviewer ML2d,
>
> We sincerely appreciate your time and effort in providing such meticulous reviews and insightful comments.
>
> Could you please kindly let us know if our rebuttal has addressed your concerns?
>
> If you have any further questions regarding our work, we would be delighted to address them.
>
> Best regards,
>
> Authors

---

### Meta-Review · Area_Chair_jWvK · 2026-01-07

**Summary:**

The paper proposes "CapRL," a framework for improving dense image captioning using Reinforcement Learning with Verifiable Rewards (RLVR). Instead of relying on subjective human preferences or potentially biased model-based judges, the authors define caption quality by its "utility": a good caption is one that enables a blind LLM to correctly answer visual question-answering (VQA) tasks. This reward signal is used to fine-tune a multimodal model (Qwen2.5-VL-3B). The resulting model is then used to synthesize a large-scale caption dataset. Experiments demonstrate that pretraining on this dataset leads to consistent improvements.

**Reviewer Concerns:**

Reviewer Concerns addressed by the rebuttal:

Validity of Proxy Reward (Reviewers ML2d, MPnM): The concern that optimizing for "QA utility" might encourage the model to "game" the system with unnatural captions was addressed by a double-blind human study, where CapRL-3B outperformed the baseline in "Expression Fluency" (4.65 vs 4.55) and "Detail Coverage".

Fairness of Evaluation Methodology (Reviewer UVsw): The concern regarding the disjoint caption-only evaluation was addressed by pointing to the standard pre-training evaluation in Tables 1 and 2, where models pre-trained on CapRL data consistently outperformed baselines (ShareGPT4V, DenseFusion) on 12 downstream benchmarks.

Hallucination Metrics (Reviewer UVsw): The lack of quantitative evidence for hallucination reduction was resolved by providing POPE scores, where CapRL-5M surpassed baselines like Share-1M.

Reviewer Concerns that remains:

Application to Abstract Domains (Reviewer hugc): While the authors proposed qualitative strategies for handling non-factual images via style-based QA, experimental validation for these creative domains remains an area for future work.

**Reviewer Scores:**

**Reviewer ML2d (Score: 4 -> 6):** This reviewer’s fundamental concern was that the proxy reward might incentivize "gaming" the metric rather than producing natural captions.

**Reviewer MPnM (Score: 4 -> 6):** Similar to ML2d, this reviewer questioned the validity of the "utility" proxy and requested standard caption metrics. The authors added ALOHA, CLAIR, and human evaluations demonstrating the method's superiority in quality and factuality.

**Reviewer UVsw (Score: 6 -> 6):** The review will keep a positive score.

**Reviewer hugc (Score: 4 -> 6):** The primary concerns were the "heavy computational overhead" of the pipeline and sensitivity to the reward model. The rebuttal provided a runtime analysis showing a minor increase in training time and demonstrated robustness across different judge model sizes.

---

### Decision · Program_Chairs · 2026-01-26

Accept (Poster)